# Do Barrier Test Results Predict Survival in Specialist Police Tactical Selection Courses?

**DOI:** 10.3390/ijerph16183319

**Published:** 2019-09-09

**Authors:** Jeremy Robinson, Ben Schram, Elisa Canetti, Robin Orr

**Affiliations:** 1Specialist Response Group—Tactical Response Team, Australian Federal Police, Canberra 2601, Australia; 2Faculty of Health Science and Medicine, Bond University, Robina 4226, Australia (E.C.) (R.O.); 3Tactical Research Unit, Bond University, Robina 4226, Australia

**Keywords:** SWAT, special forces, barrier testing, tactical, law enforcement, strength, agility, aerobic capacity

## Abstract

Entry to specialist police tactical teams is governed by performance on a physically intense and psychologically demanding selection course. The aim of this study was to determine the attributes associated with completion of a specialist police selection course. Data pertaining to 18 candidates was obtained including 1 min push-ups, loaded pull-ups, loaded 30 m crawl, agility run, 1.2 km run and multi-stage fitness assessment. Comparisons from those who did and did not complete the selection week were performed and a hierarchical multiple regression performed. Eleven candidates finished, with significant difference found in those who completed the course in push-ups (+9.1 reps), loaded pull-ups (+2.9 reps), 1.2 km run (−16 s), loaded crawl (−6.3 s), agility (−0.67 s) and VO_2max_ (+4.8 mL/kg/min). In combination, the fitness assessments pull-ups, 30 m loaded crawl and agility time were found to predict 70% of the variability in course completion (adjusted R^2^ = 0.70, *F* (3,14) = 14.373, *p* = 0.001). When assessed independently, push-ups, 1.2 km run and VO_2max_ results only predicted a non-significant 0.02%, 0.29% and 0.12%, respectively, of course completion. Completion was influenced by aerobic fitness, upper limb strength and endurance and agility. These variables appear to be predictive of course success.

## 1. Introduction

To enter special police tactical teams, candidates must pass an initial barrier assessment before attempting a selection course. Barrier testing is typically comprised of a battery of physical assessments designed to ensure candidates are capable of completing the selection course and do not have an elevated risk of injury [1]. Only after acceptable performance on a barrier test is a candidate able to attempt a selection course. Selection courses themselves are designed to be physically intense and psychologically demanding, to match the operational tempo of specialist tactical policing [2]. Examples of tasks conducted during selection courses may include periods of sleep deprivation with loaded pack marches, repeated carriage of stores and stimulus response scenarios with armed offenders. 

Given that specialist tactical police are tasked with load carriage, above and beyond that of general police, a high level of strength, endurance and aerobic fitness is required [3]. A recent review highlighted the importance of these key areas within elite tactical units, with their relationship not only to occupational task performance, but also to injury risk [4]. It is, therefore, assumed that a candidate for selection within tactical units must possess a high level of fitness in a multitude of areas. 

To date, minimal research exists pertaining to specialist tactical populations, or specifics regarding the physical attributes associated with success in selection courses. It is clear that these populations must have a high level of a variety of fitness attributes as highlighted in previous research [4]. Therefore, despite the importance of assessing a variety of fitness attributes within this population, it is unclear which, if any, are associated with or predictive of survival in a demanding selection week. A greater understanding of the physical attributes of success in a selection course could ensure the capture of appropriate candidates, reduce attrition rates, decrease injury risk and increase the efficiency of the selection process [1]. The lack of research in this area creates a gap in the knowledge base in this area and consequently, the aim of this study was to determine the fitness attributes associated with completion of a weeklong, specialist police tactical selection course. 

## 2. Materials and Methods

A retrospective cohort study where deidentified data pertaining to barrier testing physical assessment results were obtained from a specialist police tactical unit was carried out. The result of the subsequent selection course—being either completion or non-completion—was also recorded. Candidates could be removed at any point of time on the selection course by either the specialist police tactical directional staff or medically withdrawn after consultation with the course Paramedics and specialist medical staff or withdraw on their own accord. The selection course was conducted over a five-day period, the location of which was not provided for security reasons. 

Barrier testing data included measures of age, height and weight along with the following physical assessments: 

*Grip Strength:* Grip strength was measured on the dominant and non-dominant hand using a Jamar dynamometer (Jamar, Sammons, Preston, IL, USA). The handle was adjusted to fit between the palm of the hand to the middle of the four fingers as per the protocol by Dortkamph [5]. The arm was maintained by the side with the elbow in full extension prior to candidates being instructed to squeeze and maintain a smooth maximal contraction for 5 s. The best of two trials was recorded unless the difference in results was greater than 5%, in which case, a third trial was undertaken. Results were recorded to the nearest 100 g. 

*7-Stage Abdominal Strength:* Strength of the abdominals was measured with a phased 7-stage assessment which also followed the protocol outlined by Dortkamph [5]. This assessment required completing a single sit-up with progressive difficulty from body weight through the final stage with a 5 lb weight behind the head. The level which the candidate completed was recorded as their score for this assessment. 

*Push-ups:* The push-up assessment was the results of the maximum number of full push-ups completed within a 60 s period. Candidates were required to perform a complete repetition from full elbow extension to 90 degrees of elbow flexion. The push-ups assessment is a measure of the muscular endurance capacity of the upper limb. 

*Multi-stage Fitness Assessment:* A multi-stage fitness assessment or beep test as it is commonly known was conducted to measure candidate’s aerobic fitness. A 20 m distance was run between at a gradually increasing pace to an audio cue [6]. The final stage and shuttle number was recorded with the results converted to a relative VO_2max_ (mL/kg/min) for each candidate as described by Leger [6]. The test was conducted on a flat, hard rubber, non-slip surface inside a gymnasium with a 20 m distance between two identifiable cones marked out with a measuring tape (30 m Fibre Glass Hart Sport). The beep test intervals were governed by an audio compact disc from the Australian Sports Commission with each level progressively increasing in speed. Candidates were instructed that the test was an ‘individual maximal aerobic power running test’ and that they were required to run towards the opposite 20 m line and reach that line before the next successive beep (preferably in time with the successive beep). This constituted completion of a successful ‘shuttle’. They were then required to return to the original line within the sound of the following beep. This same pattern was to be performed continuously, with successive increases in speed between levels until they reach voluntarily exhaustion. If the candidates failed to reach the opposite line before the ‘beep’, the candidates were issued with one fail attempt. If they recorded two consecutive fail attempts, they were withdrawn from the test and their score (Level and shuttle number) were recorded. However, if the candidates reached the next line before the second consecutive beep, their fail attempts were reset.

*1.2 km Run*: A run over a distance of 1.2 km was completed by candidates as quickly as possible. Candidates wore physical training gear with no external load. The time was measured to the nearest second. 

*Agility:* The Illinois Agility test was conducted as per previous investigations [7]. A track was measured out and marked by cones on a hard no-slip surface and the candidate’s time to completion was recorded to the nearest 0.01 s.

*Loaded Pull-ups:* The maximum number of pull-ups completed with an additional 17 kg of load in a Sord brand plate carrier vest was recorded. The pull-ups were conducted with a pronation grip (back of hands facing away from the officer) with a grip-width wider than shoulder width in order to allow a 90-degree angle when upper arms were parallel to the ground. Legs were bent at the knee at 90 degrees with ankles crossed over behind. No swinging of the legs was allowed during the pull-up movement. On raising up, the candidates chin was required to be raised above the level of the bar. Raising to this position constituted a successful repetition.

*Method of Entry (MOE):* An occupational tactical specific task was designed whereby candidates began in a stairwell at the bottom of 3 flights of stairs. Once the scenario began, candidates preceded to run up the stairs wearing police specialist overalls, boots, 17 kg Sord ballistic vest and carrying a 10 kg sledgehammer in one hand and a 12 kg ballistic shield in the other. Time was recorded to the nearest 0.01 s with a Hart Sports handheld timer by an Strength and Conditioning (S&C) coach. 

*400 m Swim:* Candidates wore swimming attire only (not police specialist overalls) and were timed for the duration in which it took them to complete a 400 m swim in a 50 m pool. The candidates were only permitted to complete the swim using a forward-facing freestyle stroke in a continuous movement without the use of goggles. The candidates were not permitted to stop at any time, including holding on to the wall at either end of the 50 m pool. Candidates were permitted to touch the wall at each end with either their hand or feet before recommencing the swim. Time to complete the swim was measured to the nearest 0.01 s with a Hart Sports handheld timer by an S&C coach.

*Dummy Drag:* Candidates were timed while completing an 80 kg dummy drag whilst wearing a 17 kg Sord plate carrier vest. The task involved dragging the dummy as far as possible for 10 s with 20 s rest, repeated six times. The distance covered was measured to the nearest 5 m and taken as the result. The total distance of the dummy drag course was marked out 60 m long with a cone at the end for a turnaround point. 

*30 m Loaded Crawl:* Candidates crawled on a flat grassed surface for 30 m while wearing overalls, boots and their 17 kg Sord brand plate carrier. The time to complete the distance was measured to the nearest 0.01 s with a Hart Sports handheld timer by an S&C coach.

*Pack March:* A pre-defined route of 10 km was completed by candidates with a 25 kg pack. The dress for the test was issued overalls and boots. Each of the candidates’ individual operational backpacks were weighed (Wedderburn Ds-530 Digital Industrial scale) to ensure a load of 25 kg. The primary weapon carried was unloaded and weighed 3.6 kg. The 10 km pack march route was conducted on a primarily flat surface with areas of slight gradient that was a combination of bitumen and hard dirt. The route was marked out using a Garmin Oregon 600 t handheld GPS. The route undertaken for the test was not provided due to security reasons for operational training location. The candidates were instructed that they could shuffle during the test when required as this may be expected during tactical movements operationally. The time to complete the 10 km was recorded to the nearest 0.01 s on a Hart Sports handheld timer by an S&C coach. 

### Statistical Analyses

Data were received in excel spreadsheet form before being transferred to the Statistical Package for the Social Sciences software program (SPSS, Version 24, IBM, Armonk, NY, USA). Variables were analysed with an independent samples t-test to determine whether there were any differences between those who did and did not complete the selection week. A hierarchical multiple regression was performed to determine a model which could predict course success using these variables. The overall level of significance was set at 0.05 a priori. 

## 3. Results

From an initial 18 candidates who attempted the selection course, 11 completed the selection week, with two self-withdrawing, and five withdrawn medically. The barrier results of those who did and did not complete the week are seen in Table 1 below. Those who completed the course were, on average, younger and shorter than those who did not complete the course; however, these differences were not significant. There was a significant difference in weight in those who completed the course being, on average, 10.5 kg (95% CI = 3.4–17.6) lighter than those who did not complete the course. There were no significant differences with respect to grip strength on either hand, abdominal strength, MOE time, 400 m swim, dummy drag or pack march. Significant differences were seen in push-ups and pull-ups, with those who completed the course completing, on average, 9.1 more repetitions of push-ups (95% CI = 2.6–15.6, t(16) = 2.989, *p* = 0.009) and 2.9 more repetitions of the loaded pull-up (95% CI = 1.3–4.6, t(16) = 3.77, *p* = 0.002). Those who completed the week also possessed a 1.35 mL/kg/min better VO_2max_ (95% CI = 0.18–2.51, t(16) = 2.455, *p* = 0.026), a 32 s quicker 1.2 km run time (95% CI = 0.09–0.53, t(16) = 3.069, *p* = 0.007), were 6.3 s quicker in the loaded crawl task (95% CI = 2.70–9.91, t(16) = 3.708, *p* = 0.002) and 0.67 s quicker in the agility run (95% CI = 0.25–1.09, t(16) = 3.387, *p* = 0.02).

A hierarchal linear regression was performed which found that agility, the 1.2 km run, loaded crawl and pull-ups created a significant model to predict course completion (adjusted R^2^ = 0.68, *F* (4,13) = 10.126, *p* = 0.001); however, a strong correlation was found between the 1.2 km run time and the agility time (r = 0.788) and the subsequent removal of the 1.2 km run led to an increase in the predictive ability of the model (adjusted R^2^ = 0.70, *F* (3,14) = 14.373, *p* = 0.001). The results of the pull-ups, agility and loaded crawl could predict 70% of the variability in course completion. Despite showing significant differences in those who did and did not complete the course, as independent predictors, push-ups—which only explained 0.02% of the variance—the 1.2 km run (0.29%) and VO_2max_ results (0.12%) did not have a significant influence on course completion.

## 4. Discussion

The aim of this study was to determine the fitness attributes associated with completion of a weeklong, specialist police selection course. The results of this study suggest that candidates who are successful in completing a physically intense selection specialist tactical police selection course, are lighter, can complete more push-ups, possess greater aerobic fitness, are more agile, can complete more loaded pull-ups and can complete a loaded crawl quicker than those who did not complete the course. Further analysis revealed that performance in loaded pull-ups, agility and the loaded crawl in particular, were associated with course completion. The assessments which highlighted differences in groups are associated with occupational performance within this population. Load carriage, being able to move quickly and for long periods is an important attribute in specialist tactical populations [3,4]. These variables analysed had high fidelity to the individual events of the selection course, e.g., loaded pack march, the carriage of stores and injured officer recovery drills. This study highlights the importance of upper limb strength, the ability to move quickly, and move under load as important for high performance and survival of a selection course. 

It should be noted that the results in the other assessments may exhibit a ceiling effect, and, therefore, should not be dismissed as unimportant within this population. For example, previous studies in military special forces selection candidates have found that aerobic fitness was not associated with selection as the overall fitness level was very high [1]. Despite a significant difference being found between those who did and did not complete the course in this study, aerobic fitness did not feature in the predictive modelling. A similar argument could be made for the push-ups assessment which has been shown to be related to selection within specialist police units in previous investigations [2]. The results of the push-up assessment is considered to be ‘excellent’ when compared to normative values [8], and compare well to previous reports of 58.8 ± 11.9 in Australian specialist police [2], 64.5 ± 14.1 in the United States specialist police and 69 ± 12 in special forces soldiers [1].

Despite not showing any differences between groups, the results of the grip strength assessment are of excellent standard when compared to normative data [9]. Grip strength has been shown to be important within tactical populations due to its relationship to occupational performance within this population [10]. Likewise, while few studies exist with comparable data for this population, the results reported for the agility assessment are much quicker than the 21.04 ± 2.57 reported among general police officers [11]. The loaded pull-up assessment featured as a point of difference between groups and strongly in the predictive model. While minimal comparative data exists, the numbers reported here compare well to a cohort of Navy Special Operation Solders who performed 9.0 ± 4.0, albeit with a lighter weight of 11.25 kg.

This study is limited by the small sample size available, despite recruiting the entire course. The interim results of this study may inform future studies of a greater size. Likewise, a ceiling effect of many of the fitness measures may have existed, and, therefore, the results may differ in subsequent courses. 

## 5. Conclusions

It appears as though a high level of fitness in a variety of areas including aerobic fitness, muscular strength and agility are important for survival of a physically intense specialist police selection course. Of note, is the contribution of a loaded crawl and loaded pull-up to survival, both of which are deemed to be occupationally relevant. This study highlights the importance of a multitude of fitness attributes in this population, with aerobic fitness not being predictive of course success; however, this may be due to a ceiling effect. Strength and Conditioning Coaches and Health professionals working with candidates or individuals planning on attending selection courses should aim to maximize upper limb strength, agility and the ability to move under load to maximize chances of course completion. 

## Figures and Tables

**Table 1 ijerph-16-03319-t001:** A comparison of 18 police candidates who did and did not complete a specialist selection course. Results reported as mean ± SD.

Measure	Completed (*n* = 11)	Not Completed (*n* = 7)	*p* Value
Age (years)	30.6 ± 5.0	34.4 ± 4.5	0.123
Height (cm)	182.8 ± 4.9	185.1 ± 7.2	0.423
Weight (kg)	85.4 ± 6.7	95.9 ± 7.4	0.007
Grip Strength Dominant (kg)	65.4 ± 6.3	60.6 ± 4.8	0.106
Grip Strength non-dominant (kg)	61.2 ± 6.2	57.5 ± 7.4	0.265
Abdominal Strength (1–7)	7.0 ± 0.0	6.4 ± 0.8	0.103
Push-ups (reps)	63.8 ± 6.8	54.7 ± 5.3	0.009
VO_2max_ (mL/kg/min)	53.7 ± 2.3	48.9 ± 4.9	0.013
1.2 km Run (m:s)	4:25 ± 0.09	4:41 ± 0.21	0.002
Agility (s)	15.7 ± 0.3	16.3 ± 0.6	0.004
Pull-ups (reps)	7.5 ± 1.7	4.6 ± 1.5	0.002
MOE stair well run (s)	27.7 ± 3.7	30.2 ± 5.2	0.254
400 m Swim (m:s)	9:05 ± 1:33	9:25 ± 1:31	0.674
Dummy Drag (s)	97.7 ± 6.5	97.1 ± 9.5	0.878
Loaded Crawl (s)	26.8 ± 3.4	33.1 ± 3.7	0.002
10 km Pack March (m:s)	1:26.33 ± 2.12	1:28.11 ± 2.04	0.139

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
