# Peer review of "Do Barrier Test Results Predict Survival in Specialist Police Tactical Selection Courses?"

_ijerph, 2019, doi:10.3390/ijerph16183319_

Round 1
Reviewer 1 Report
1- The introduction is weak and does not have a scientific research gap;
2- The discussion can be improved by explaining how the studied variables interfered with performance
Author Response
Reviewer 1:
1- The introduction is weak and does not have a scientific research gap;
Thank you for your comment and for your time in reviewing our manuscript. There is minimal information available in the scientific literature in this area and therefore it remains difficult to discuss many previous studies in the introduction. We do acknowledge your concern however and therefore we have added to the introduction, highlighting the research gap which is centred around the minimal research in this area.
2- The discussion can be improved by explaining how the studied variables interfered with performance
Thank you for your comment. We have elaborated in the discussion how these specific variables would have assisted in each event during the selection course.
Reviewer 2 Report
First, I want to thank the authors for their continued work with our first responders and tactical units. This work is needed to help everyone. Overall, I think this is a very well written article with some interesting findings. I do have some minor edits, questions, and suggestions listed below.
Page 1 Lines 21-24: These three sentences seem to contradict or conflict with each other. The first sentence says there is no run and VO2 Max did not have influence on completion. Then the next sentence says aerobic fitness influences completion. There needs to be clarification here.
Page 1 Line 39: Authors might want to change the phrase “will possess” to “must possess” as this helps the reader understand that it is a necessity to the job.
Page 1 Introduction: Might be help if examples of the tasks in the selection course were presented somewhere to give reader a better understanding of the tasks that need to be completed.
Page 2 Lines 63-67: Just need a little clarification, does the candidate complete each stage with just one sit-up?
Page 2 Line 75: Explain how conversion of Beep Test was made to relative VO2 max. Was this a proprietary equation or was extra data entered?
Page 2 Lines 89-91: was the run performed on track or another surface?
Page 3 Line 115: the sentence is unclear especially the phrase “dummy drag six times over 10 seconds.” Maybe try to reword sentence
Page 4 Line 148: Did not see MOE defined earlier in paper. Please define earlier to use it here.
Page 4 Line 148 Add a “S” to “difference” to make it “significant differences”
Page 4 Table Title: A table should stand on its own and the title should give a clear representation of what the data is representing. The present title doesn’t mention who is completing the selection course or what the selection course is. It also doesn’t let the reader know what the numbers are that have been presented. A suggested title could be similar to “A comparison of 18 Police Specialist Candidates who did or did not complete the Tactical Selection Course (mean ± SD).”
Page 4 Table Measure Column: On some measures (m:s) is used and others (min:sec) are used. Please be consistent.
Page 4 Table Values: Please put a space between the number and the ± symbol and the next number. Like this 30.6 ± 5.0. It is a little harder to read in its current format.
Page 4 Lines 163-165: You all might want to report the predictive ability of a model with the three variables listed in this sentence as they did have significant differences. Just a suggestion, as it was an interesting find.
Page 4: Line 170: Leaner would imply body composition analysis was performed, but that was never reported. Lighter might be a better term.
Page 5: Line 224: Reference 4 has ORR and SCHRAM in all caps and should be fixed.
Author Response
Reviewer 2:
First, I want to thank the authors for their continued work with our first responders and tactical units. This work is needed to help everyone. Overall, I think this is a very well written article with some interesting findings. I do have some minor edits, questions, and suggestions listed below.
Thank you for your comments. Please find your specific comments addressed below.
Page 1 Lines 21-24: These three sentences seem to contradict or conflict with each other. The first sentence says there is no run and VO2 Max did not have influence on completion. Then the next sentence says aerobic fitness influences completion. There needs to be clarification here.
Apologies for the confusion, the constraints on the word count in the abstract made it difficult to explain. For clarity, there were significant differences in push-ups, VO2max, 1.2km run, agility and pull ups between groups. The regression model featured pullups, the crawl and agility, however when examined independently as part of the regression model, the push-ups, 1.2km run and VO2max did not influence completion. This has been changed in the abstract to read In combination, the fitness assessments pullups, 30m loaded crawl and agility time were found to predict 70% of the variability in course completion (adjusted R2=0.70, F (3,14) = 14.373, p= 0.001). When assessed independently, push-ups, 1.2km run and VO2max results only predicted a non-significant 0.02%, 0.29% and 0.12%, respectively, of course completion.
The results section elaborates on this where it reads:
A hierarchal linear regression was performed which found that agility, the 1.2km run, loaded crawl and pullups created a significant model to predict course completion (adjusted R2=0.68, F (4,13) = 10.126, p= 0.001), however a strong correlation was found between the 1.2km run time and the agility time (r=0.788) and the subsequent removal of the 1.2km run led to an increase in the predictive ability of the model (adjusted R2=0.70, F (3,14) = 14.373, p= 0.001). The results of the pullups, agility and loaded crawl could predict 70% of the variability in course completion. Despite showing significant differences in those who did and did not complete the course, as independent predictors, pushups which only explained 0.02% of the variance, the 1.2km run (0.29%) and VO2max results (0.12%) did not have a significant influence on course completion.
Page 1 Line 39: Authors might want to change the phrase “will possess” to “must possess” as this helps the reader understand that it is a necessity to the job.
Good point, this has been changed in the manuscript.
Page 1 Introduction: Might be help if examples of the tasks in the selection course were presented somewhere to give reader a better understanding of the tasks that need to be completed.
Thank you for your comment, we have added the following for the reader: Examples of tasks conducted during selection courses may include periods of sleep deprivation with loaded pack marches, repeated carriage of stores and stimulus response scenarios with armed offenders.
Page 2 Lines 63-67: Just need a little clarification, does the candidate complete each stage with just one sit-up?
Thank you for your comment. Only a single successful completion is required to progress to the next stage, this has been added to the manuscript.
Page 2 Line 75: Explain how conversion of Beep Test was made to relative VO2 max. Was this a proprietary equation or was extra data entered?
The equation by Leger was used to convert these scores. This has been added to the manuscript for clarity.
Page 2 Lines 89-91: was the run performed on track or another surface?
Thank you for your comment, we have added some more detail regarding this surface for this assessment.
Page 3 Line 115: the sentence is unclear especially the phrase “dummy drag six times over 10 seconds.” Maybe try to reword sentence
Thank you for your suggestion, this has been reworded in the manuscript.
Page 4 Line 148: Did not see MOE defined earlier in paper. Please define earlier to use it here.
Apologies, we have added the abbreviation to the Method of Entry part of the methods to define it the first time.
Page 4 Line 148 Add a “S” to “difference” to make it “significant differences”
Thank you, this has been added.
Page 4 Table Title: A table should stand on its own and the title should give a clear representation of what the data is representing. The present title doesn’t mention who is completing the selection course or what the selection course is. It also doesn’t let the reader know what the numbers are that have been presented. A suggested title could be similar to “A comparison of 18 Police Specialist Candidates who did or did not complete the Tactical Selection Course (mean ± SD).”
Thank you for your comment, the title has been changed to read A comparison of 18 police candidates who did and did not complete a specialist selection course. Results reported as mean±SD.
Page 4 Table Measure Column: On some measures (m:s) is used and others (min:sec) are used. Please be consistent.
Thank you for your comment, this has been changed for consistency.
Page 4 Table Values: Please put a space between the number and the ± symbol and the next number. Like this 30.6 ± 5.0. It is a little harder to read in its current format.
Thank you for your suggestion, we have made this change to the table.
Page 4 Lines 163-165: You all might want to report the predictive ability of a model with the three variables listed in this sentence as they did have significant differences. Just a suggestion, as it was an interesting find.
Thank you for your comment, the two models, with 5 and 3 variables respectively, have been described in the last paragraph of the results section. Please see below for your convenience.
A hierarchal linear regression was performed which found that agility, the 1.2km run, loaded crawl and pullups created a significant model to predict course completion (adjusted R2=0.68, F (4,13) = 10.126, p= 0.001), however a strong correlation was found between the 1.2km run time and the agility time (r=0.788) and the subsequent removal of the 1.2km run led to an increase in the predictive ability of the model (adjusted R2=0.70, F (3,14) = 14.373, p= 0.001).
Page 4: Line 170: Leaner would imply body composition analysis was performed, but that was never reported. Lighter might be a better term.
Thank you, this has been changed to ‘lighter’ in the manuscript.
Page 5: Line 224: Reference 4 has ORR and SCHRAM in all caps and should be fixed.
Thank you, this has been changed in the manuscript.